# Structure–Activity Relationships of Low Molecular Weight Alginate Oligosaccharide Therapy against *Pseudomonas aeruginosa*

**DOI:** 10.3390/biom13091366

**Published:** 2023-09-08

**Authors:** Manon F. Pritchard, Lydia C. Powell, Jennifer Y. M. Adams, Georgina Menzies, Saira Khan, Anne Tøndervik, Håvard Sletta, Olav Aarstad, Gudmund Skjåk-Bræk, Stephen McKenna, Niklaas J. Buurma, Damian J. J. Farnell, Philip D. Rye, Katja E. Hill, David W. Thomas

**Affiliations:** 1Advanced Therapies Group, School of Dentistry, Cardiff University, Cardiff CF14 4XY, UK; l.c.powell@swansea.ac.uk (L.C.P.); adamsj23@cardiff.ac.uk (J.Y.M.A.); khans17@cardiff.ac.uk (S.K.); steve_mckenna@outlook.com (S.M.); farnelld@cardiff.ac.uk (D.J.J.F.); hillke1@cardiff.ac.uk (K.E.H.); thomasdw2@cardiff.ac.uk (D.W.T.); 2Microbiology and Infectious Disease Group, Swansea University Medical School, Swansea SA2 8PP, UK; 3School of Biosciences, Cardiff University, Cardiff CF10 3AX, UK; menziesg@cardiff.ac.uk; 4Department of Bioprocess Technology, SINTEF Materials and Chemistry, N-7465 Trondheim, Norway; anne.tondervik@sintef.no (A.T.); havard.sletta@sintef.no (H.S.); 5Department of Biotechnology, Norwegian University of Science and Technology, N-7491 Trondheim, Norway; olav.a.aarstad@ntnu.no (O.A.); gudmund.skjak-brak@ntnu.no (G.S.-B.); 6Physical Organic Chemistry Centre, School of Chemistry, Cardiff University, Cardiff CF10 3AT, UK; buurma@cardiff.ac.uk; 7AlgiPharma AS, Industriveien 33, N-1337 Sandvika, Norway; phil.rye@algipharma.com

**Keywords:** alginate oligosaccharide, mannuronic acid, guluronic acid, *Pseudomonas aeruginosa*, biofilm, calcium

## Abstract

Low molecular weight alginate oligosaccharides have been shown to exhibit anti-microbial activity against a range of multi-drug resistant bacteria, including *Pseudomonas aeruginosa*. Previous studies suggested that the disruption of calcium (Ca^2+^)–DNA binding within bacterial biofilms and dysregulation of quorum sensing (QS) were key factors in these observed effects. To further investigate the contribution of Ca^2+^ binding, G-block (OligoG) and M-block alginate oligosaccharides (OligoM) with comparable average size DPn 19 but contrasting Ca^2+^ binding properties were prepared. Fourier-transform infrared spectroscopy demonstrated prolonged binding of alginate oligosaccharides to the pseudomonal cell membrane even after hydrodynamic shear treatment. Molecular dynamics simulations and isothermal titration calorimetry revealed that OligoG exhibited stronger interactions with bacterial LPS than OligoM, although this difference was not mirrored by differential reductions in bacterial growth. While confocal laser scanning microscopy showed that both agents demonstrated similar dose-dependent reductions in biofilm formation, OligoG exhibited a stronger QS inhibitory effect and increased potentiation of the antibiotic azithromycin in minimum inhibitory concentration and biofilm assays. This study demonstrates that the anti-microbial effects of alginate oligosaccharides are not purely influenced by Ca^2+^-dependent processes but also by electrostatic interactions that are common to both G-block and M-block structures.

## 1. Introduction

Alginates are linear anionic polysaccharides derived from brown seaweeds such as *Laminaria hyperborea* or bacteria such as *Pseudomonas fluorescens* and *Azotobacter* sp. [1]. Alginates are composed of β-D-mannuronic acid (M) and its C-5 epimer α-L-guluronic acid (G), varying in length (degree of polymerization, DPn), composition (G- or M-residues), and secondary structure (alternating G:M residues or homogenous block structures) [2]. These features directly influence the physicochemical properties of alginates and their affinity for monovalent and divalent cations such as calcium (Ca^2+^) [3,4].

High molecular weight (MW) polymeric alginates have a long history of use in a wide variety of food and biomedical applications, having a global market value projected to be worth USD 547.2 million by 2027 [5,6,7,8,9]. Recently, low MW alginate oligosaccharides, containing an average of 2–25 monomers with shorter chain lengths obtained by hydrolysis of glycosidic bonds or bio-/organic synthesis, have shown altered bioactivity and demonstrated potential health benefits [1,6]. These low MW alginate oligosaccharides retain most of the chemical and physical properties of the higher MW commercial alginates, preserving their affinity towards monovalent and divalent ions, which is dependent on chain length and composition. However, due to their low MW, they normally do not form gels in the presence of divalent cations and have both high water solubility and low viscosity, retaining their negative charge and, therefore, their affinity for divalent cations [1,10].

The inexorable increase in multi-drug resistant pathogens worldwide is of major concern, with an estimated 10 million deaths annually predicted by 2050, at a global projected cost of >USD 100 trillion [11]. Low MW alginate oligosaccharides with a high G-content have previously demonstrated anti-microbial properties against both bacterial and fungal biofilms, including those of the highly virulent/multi-drug resistant (MDR) ESKAPE pathogens against which few, if any, antibiotics are still effective [12,13,14,15,16]. High MW M-rich alginate polymers are a key constituent in *Pseudomonas aeruginosa* biofilms, with Ca^2+^ cross-linking playing an important role in biofilm stability and antibiotic resistance. Therefore, the ability of low MW alginate oligosaccharides with a high G-content to scavenge calcium has been proposed as a creditable mechanism of action in their demonstrated anti-microbial and antibiofilm activity [17]. These studies implicate monomer composition and distribution (M:G ratio) in these activities.

A clear understanding of low MW alginate oligosaccharide-related structure–activity relationships is necessary to determine their mechanism of action and facilitate the development of targeted therapies/precision medicines for clinical application. Here, using engineered G- and M-block alginates (of comparable DPn and MW distribution), we sought to determine the role of Ca^2+^ binding on the previously observed antibacterial and antibiofilm effects of the G-block alginates against the respiratory human pathogen *P. aeruginosa*. Furthermore, the extent to which the structure–activity relationships of G- and M-blocks influence the anti-microbial potential of these low MW alginates was also investigated.

## 2. Materials and Methods

### 2.1. Production and Characterization of Low MW Alginate Oligosaccharides

Low MW alginate oligosaccharides with a high G-block and M-block content were prepared with comparable number averaged DP values. The preparation of alginate fractions is summarized in detail elsewhere [18]. Briefly, both alginate variants were derived from different sources: high G content variants being derived from *Laminaria hyperborea* stipe preparations following acid hydrolysis to give a G-rich polydisperse mixture of polymers, while high M content poly-mannuronic acid variants were produced by an *AlgG* negative strain of *P. fluorescence*. Bacterial alginates were deacetylated by mild alkaline treatment. Poly-mannuronic variants were depolymerized by the two-step acid hydrolysis described previously [19]. Fractions were analyzed by high-performance anion-exchange chromatography with pulsed amperometric detection (HPAEC-PAD; Dionex ICS-5000, Sunnyvale, CA, USA). Metal ion concentrations were determined by ICP analysis (Inductively Coupled Plasma Atomic Emission Spectroscopy, ICP-AES) to determine concentrations of trace elements in the various preparations. OligoG and OligoM with an average DPn of 19 were used in all in vitro experiments.

### 2.2. Bacterial Strains and Culture Media

*Pseudomonas aeruginosa* PAO1 (ATCC 15692) and a mucoid cystic fibrosis isolate NH57388A from a patient attending the Danish CF Centre, Rigshospitalet, Copenhagen, [20] were utilized throughout the study. The strains were sub-cultured on blood agar base no. 2 (BA; LabM, Lancashire, UK) supplemented with 5% horse blood and overnight cultures were grown at 37 °C in Tryptone Soy Broth (TSB; Oxoid, Hampshire, UK). Experiments were carried out in Mueller–Hinton (MH; LabM, Lancashire, UK) broth. *Chromobacterium violaceum* cultures (ATCC 31532 and CV026) in Luria–Bertani (LB; Lab M, Lancashire, UK) broth were used for the quorum sensing (QS) studies [21].

### 2.3. Fourier-Transform Infrared Spectroscopy (FTIR)

Overnight cultures of *P. aeruginosa* (PAO1) were centrifuged at 3000× *g* for 20 min and washed (×2) in dH_2_O. Cultures were standardized to an optical density at 600 nm (OD_600_) of 1 before being re-centrifuged at 3000× *g* for 20 min and resuspended in 66.6 µL dH_2_O ± 0.5% OligoG or 0.5% OligoM. The samples were then incubated at room temperature for 20 min before being subjected to hydrodynamic shear (5000× *g*; 6 min) to remove excess alginate oligosaccharide. The supernatant was removed, and the pellet was resuspended in dH_2_O prior to being centrifuged (5000× *g*, 6 min) and resuspended in dH_2_O. The sample (15 μL) was pipetted onto the attenuated total reflectance (ATR) sampling module of a Nicolet 380 FTIR Instrument (Thermo Fisher Scientific, Loughborough, UK) and allowed to air dry prior to recording the IR spectra. Wavelength scans were taken between 4000 and 525 cm^−1^ at a resolution of 4 cm^−1^ and 64 scans taken to create a mean. Each sample measurement was determined from (n = 6) biological replicates and (n = 3) technical replicates

Data processing and analysis were performed using the R statistical environment [22]; wavenumbers from 1850 to 950 cm^−1^ were pre-processed using two-point liner subtraction baseline correction methods and then normalized using vector normalization. The Wilcox Rank Sum test was used to determine wavenumbers that were significantly different (*p* < 0.05) between the OligoG and OligoM spectra. To compensate for multiple testing, Bonferroni correction was applied. These selected wavenumbers were put forward into hierarchical cluster analysis (HCA), and a dendrogram produced.

### 2.4. Molecular Dynamics (MD) Simulations

MD simulations were performed on a Gram-negative lipopolysaccharide (LPS-DPPE) lipid layer [23] to mimic the common wound pathogen *P. aeruginosa* PAO1 bacterial outer membrane with/without G-block (degree of polymerization; DP 6) or M-block (DP 6) alginates composed of six monomers [24]. Simulations with oligosaccharides larger than this (e.g., DPn 19) were not possible due to the exponential increase in computing time that would have been required. GROMACS software [25] was used to run MD protocols, using assisted model building with energy refinement (AMBER)-based force field parameters and thermally equilibrated coordinates reported for the LPS-DPPE membrane [26]. Force field parameters, including partial charges, for the M- and G-blocks were taken as previously described [27]. The particle mesh Ewald (PME) method was used to treat long-range electrostatic interactions, and a 1.4 nm cut-off was applied to Lennard–Jones interactions. The MD simulations were all conducted in the NPT ensemble (number of atoms [N], pressure [P], and temperature [T] conserved), with periodic boundary conditions at a temperature of 310 K and a pressure of 1 atm. Each simulation was performed using a three-step process: steepest-descent energy minimization with a tolerance of 1000 kJ^−1^ nm^−1^; a pre-MD run (PR) with 25,000 steps at 0.002 f.s. per second per step, making a total of 2500 ps and an MD stage run for a total of 300 ns. Root mean square deviation (RMSD) of the membrane and alginate block was monitored along with the total energy, pressure, and volume of the simulation to monitor the stability of the simulations. Analyses of the simulations were performed using GROMACS tools and the resulting structures visualized using VMD [28].

### 2.5. Isothermal Titration Calorimetry (ITC)

Calorimetric titrations were carried out at 37 °C on a MicroCal PEAQ-ITC microcalorimeter (Malvern Instruments Ltd., Worcestershire, UK). The instrument was operated by applying a reference power of 10 µcal/s in high-feedback mode, stirring the sample cell contents at 750 rpm, with a pre-injection initial delay of 60 s. Freshly prepared solutions, in the required buffers, of LPS (from *P. aeruginosa* 10; Sigma-Aldrich, Dorset, UK) at 10 mg/mL, i.e., ~0.5 mM, and OligoG and OligoM at 20 mg/mL, i.e., ~5.29 mM in terms of 19 mer units, were loaded into the calorimeter sample cell and injection syringe, respectively. Concentrations were estimated from weights per volume using a MW of 20 kDa for LPS (representing the non-aggregated monomeric molecular form) [29]. The MW of a 19 mer of OligoG and OligoM was estimated as follows: 216.12 g/mol sodium guluronate × 19 = 4106.28 g/mol minus 18 water molecules (324.28 g/mol) equals 3782.00 g/mol per oligomer. Buffers employed in the experiment were 100 mM NaCl and 20 mM MOPS (adjusted to pH 7 with NaOH) with 1 mM or 2.5 mM CaCl_2_. All experiments involved an initial injection of 0.4 µL in 0.8 s followed by 18 further injections of 2.0 µL in 4.0 s per injection into the calorimeter sample cell. Injections were spaced by at least 90 s to allow full recovery of the baseline. Raw data were treated using MicroCal PEAQ-ITC Analysis Software (1.0.0.1259) to generate both integrated heat effects per injection (ΔQ) and molar heat effects per injection (ΔH).

### 2.6. Bacterial Growth Curves

Overnight cultures of *P. aeruginosa* NH57388A were adjusted to 1 × 10^7^ cfu/mL in OligoG and OligoM (0–6%; *w*/*v*). Samples were grown in 96-well plates for 48 h at 37 °C, and changes in cell density (OD_600_) were recorded every hour on a FLUOstar Omega plate reader. The minimum significant difference (MSD) was calculated using the Tukey–Kramer method using Minitab 17.2.1 (Minitab Inc., State College, PA, USA).

### 2.7. Biofilm Formation Assay

Biofilms were grown in Whatman 96-well glass-bottomed plates by inoculating wells with *P. aeruginosa* NH57388A (10^6^ cfu/mL) grown for 24 h (37 °C; 120 rpm) in MH broth ± 0.5, 2, or 6% OligoG and OligoM

### 2.8. Confocal Laser Scanning Microscopy (CLSM) Imaging and Analysis

Following incubation, the biofilm supernatant was removed, and the biofilm stained with 6% (*v*/*v* in phosphate-buffered saline [PBS]) LIVE/DEAD^®^ (BacLight^TM^ Bacterial Viability Kit, Invitrogen, Thermo Fisher Scientific, Loughborough, UK) for 10 min and z-stack CLSM images were taken (Leica TCS SP5). CLSM images were analyzed using COMSTAT image analysis software for changes in bacterial biomass volume [30].

### 2.9. Quorum Sensing Assay (QS)

A small-scale QS assay measuring violacein pigment was adapted from Manner and Fallarero, 2018 [21]. *Chromobacterium violaceum* cultures (ATCC 31532 and CV026) were adjusted with LB broth to OD_600_ 0.7 (~1 × 10^9^ cfu/mL). Adjusted cultures (200 μL/well) ± OligoG or OligoM (0.5, 2, and 6%) were placed in a flat-bottomed 96-well microtiter plate with 0.5 μM N-(β-ketocaproyl)-L-homoserine lactone (Sigma–Aldrich, Dorset, UK; C6-HSL dissolved in dimethyl sulfoxide [DMSO]) to induce pigment production and incubated at 30 °C (200 rpm shaking for 48 h in darkness). Furanone (Sigma–Aldrich, Dorset, UK; 2[5*H*]-Furanone 98% dissolved in DMSO), a QS inhibitor, was used as a positive control (final concentration 1 mg/mL), inhibiting the production of violacein. A DMSO equivalent was also added in untreated control wells. To quantify the violacein pigment, plates were centrifuged (Heraeus Labofuge 400 R; Thermo Fisher Scientific, Loughborough, UK) at 2000× *g* for 10 min and the supernatant removed before adding 200 μL of 96% (*v*/*v*) ethanol to dissolve the violacein. Samples were re-centrifuged as described above. The supernatant (100 μL/well) was then transferred to a new sterile flat-bottomed 96-well microtiter plate and the absorbance (OD_595_) was read using a FLUOstar Omega plate reader. The inhibition percentage of violacein production was calculated as follows:$$\mathrm{Inhibition}\;\mathrm{percentage}\;\left(\%\right)=\frac{\left({\mathrm{untreated}\;\mathrm{control}\;\mathrm{OD}}_{595}-{\mathrm{sample}\;\mathrm{OD}}_{595}\right)}{\left({\mathrm{untreated}\;\mathrm{control}\;\mathrm{OD}}_{595}-{\mathrm{media}\;\mathrm{control}\;\mathrm{OD}}_{595}\right)}\times 100$$

Treatments were classified as highly active QS inhibitors if the inhibition percentage was ≥90%, moderately active if inhibited by 40–89%, and if less than 40% classed as inactive [21].

In parallel with the QS assay, duplicate *C. violaceum* ATCC 31532 and *C. violaceum* CV026 plates were set up in the same way without C6-HSL (but with the equivalent DMSO concentration used to replace it where appropriate) to measure the effect of OligoG and OligoM on cell viability. After 48 h incubation, the 96-well microtiter plates were centrifuged (Heraeus Labofuge 400 R; Thermo Fisher Scientific, Loughborough, UK ) at 2000× *g* for 10 min and all the supernatant was removed. Resazurin (Sigma–Aldrich. Dorset, UK; 20 μM prepared in PBS) was added and the plates (200 μL/well) were incubated in the dark for 30 min at room temperature before the fluorescence (λ_ex_ 560 nm and λ_em_ 590 nm) was read with a FLUOstar Omega plate reader. The bactericidal activity of the oligosaccharides was calculated as the inhibition percentage of viability as follows:$$\mathrm{Inhibition}\;\mathrm{percentage}\;\left(\%\right)=\frac{\left(\mathrm{untreated}\;\mathrm{control}-\mathrm{sample}\right)}{\left(\mathrm{untreated}\;\mathrm{control}-\mathrm{media}\;\mathrm{control}\right)}\times 100$$

Compounds with inhibition percentages > 40% were classified as bactericidal and excluded as being a QS inhibitor [21].

### 2.10. Synergistic Anti-Microbial Activity against Planktonic P. aeruginosa

The effect of OligoG and OligoM (0–6%) on antibiotic potentiation of the macrolide azithromycin was studied [31] using standard MIC broth microdilution assays as previously described [13].

### 2.11. Synergistic Anti-Microbial Activity against P. aeruginosa Biofilms

Biofilm formation was analyzed in the presence of a dual azithromycin/alginate oligosaccharide treatment using a Beckman Coulter Biomek NX^P^ robotic system. Overnight cultures of *P. aeruginosa* (PAO1) in TSB were adjusted to an OD_600_ of 0.005 in MH broth (LabM) before adding 30 µL of inoculum to the wells of 96-well black plates (Thermo Fisher Scientific, Loughborough, UK) containing 150 µL of MH broth with 0, 2, or 6% OligoG or OligoM and a gradient of azithromycin (Sigma–Aldrich, Dorset, UK), ranging from 0 to 512 µg/mL. Plates were incubated for 19 h at 37 °C (n = 4). The supernatant was removed, and the biofilms washed (4×) with 100 µL of PBS. PBS (100 µL) was then added to each well before addition of 100 µL of a BacTiter-Glo™ Microbial Cell Viability Assay (Promega, Madison, WI, USA) reagent. Plates were incubated at room temperature in the dark for 5 min, mixed briefly, and luminescence measured on a Molecular Devices SpectraMax Paradigm Multi-Mode Detection Platform. The MSD was calculated using the Tukey–Kramer method using Minitab 17.2.1 (Minitab Inc., State College, PA, USA).

## 3. Results

### 3.1. Characterization of Low MW Alginate Oligosaccharides

The OligoG alginate oligosaccharide had >85% monomer residues of G-units. OligoM had 100% monomer residues of M-units (Figure 1A). Purified OligoG and OligoM were characterized by high-performance anion-exchange chromatography with pulsed amperometric detection (HPAEC-PAD; Figure 1B).

### 3.2. The Effect of Alginate Oligosaccharide Composition on Bacterial Cell Membrane Binding

FTIR analysis demonstrated the presence of OligoG and OligoM on the pseudomonal cell surface following hydrodynamic shear (Figure 2A). Hierarchical cluster analysis (HCA) revealed clear clustering of the control samples compared to the alginate oligosaccharides-treated cells (Figure 2B), except for two biological repeats (one OligoG [G5] and one OligoM [M4]), which appeared to have lost the alginate oligosaccharide from the cell surface (Figure 2). Wilcox rank sum analysis highlighted 184 statistically different wavenumbers between OligoG and OligoM, therefore, further studies to dissect these potential interactions were conducted.

MD simulations of G- and M-blocks (DP 6) with the LPS-DPPE bilayer membrane of *P. aeruginosa* (measuring ~6 nm in thickness) in the presence of calcium were conducted over 300 ns. The simulations suggested that G-block alginates were attracted to the LPS bilayer within the first few nanoseconds (ns). This occurred through strong hydrogen bond interactions, thus breaking existing hydrogen bonds between sugar groups and/or preventing LPS from forming bonding with water molecules (Appendix A). These findings were aided by the strong attraction between the G residues and Ca^2+^ ions, overcoming the electrostatic repulsion between the large negative charge of both the oligomer and LPS (Figure 3). The G-block subsequently detached and re-attached to the LPS molecules several times with an increasing number of hydrogen bonds (maximum of 20 at 203 ns), simultaneously dislodging three Ca^2+^ ions from the membrane. This occurred at ~4 ns, ~80 ns, and ~150 ns (which began at 22 nm, 46 nm, and 60 nm from the G-block, respectively). Subsequently, the first two Ca^2+^ ions remained within 0.3 nm of the G-block residues. The G-block molecule penetrated the bacterial membrane approximately 1.6–1.8 nm before the simulation reached the halfway point, forming hydrogen bonds with WLL (2-(2-L-alanyl)-2-deoxy-D-galactosamine) or PO4 (phosphate) groups within the LPS molecule (Appendix A). In contrast, MD simulations with M-block alginates revealed hydrogen bonding between the oligomer and the LPS, which did not extend as far into the membrane as that observed for the G-block structures (Figure 3). The M-block also demonstrated significantly less movement/binding of Ca^2+^ ions, with one Ca^2+^ ion leaving the LPS bilayer at a more advanced stage (~100 ns).

The mean distance between the G-block or M-block alginate oligomers and the LPS-DPPE bilayer (centre of geometry [CoG] of the six blocks taken as a whole molecule to CoG of the entire bilayer) was 3.63 nm (SD 0.13) and 3.66 nm (SD 0.25), respectively. Crucially, the difference between the two alginate oligomers was highlighted by the larger standard deviation for the M-block. The M-block was found to sit largely on the surface of the LPS, with a greater range of movement back and forth from the LPS layer compared to the G-blocks. Once the first bond was formed, G-blocks held a mean of eight hydrogen bonds, whilst M-blocks held an mean of six hydrogen bonds. The attachment of the G- and M-blocks to LPS over time (250 ns) demonstrated differences in the distortion of the cell membrane surface (Appendix A). While the G-block entered the LPS in a vertical manner, penetrating the cell membrane surface and disrupting the bonds before changing position once the Ca^2+^ ions were bound, forming multiple bonds along the surface (Appendix A). The M-block, in contrast, entered at an angle; disrupting fewer Ca^2+^ ions and forming fewer bonds, which ultimately led to less surface disruption.

ITC was employed to detect the heat effects of OligoG and OligoM (20 mg/mL) dilutions in the presence of 1 mM or 2.5 mM Ca^2+^. The dilution effects of OligoG alone into buffer with 1 mM Ca^2+^ did not change significantly with increasing alginate oligosaccharide concentration in the calorimeter cell but were clearly different from the buffer-into-buffer control experiment (Figure 4). In the presence of 2.5 mM Ca^2+^, dilution of OligoG demonstrated an exothermic effect at the start of the titration; the dilution heat effects changing during the titration, suggesting that OligoG was self-aggregating, which is in line with our previous data in the presence of 5 mM Ca^2+^ [15]. These observations for OligoG are in agreement with the reported three-step interaction mechanism between Ca^2+^ and alginate, which includes critical behaviors in binding and is, therefore, very sensitive to concentration [32]. On the contrary, the lower heat effects and the absence of a change in pattern in dilution heat effects for OligoM on increasing the concentration of Ca^2+^ from 1 mM to 2.5 mM suggest less Ca^2+^ binding of OligoM and limited, if any, Ca^2+^-induced aggregation of OligoM.

Further ITC studies were conducted to establish the interaction between the alginate oligosaccharides and pseudomonal LPS in the presence of both 1 mM and 2.5 mM Ca^2+^. At both Ca^2+^ concentrations, the heat effects for titrating OligoG into LPS were clearly different from the combined heat effects for the reference experiments, again in keeping with our previous work. The interaction of OligoG with LPS was found to be more endothermic at higher calcium concentrations. These observations strongly suggested that OligoG and LPS interacted in the presence of calcium. Although the interaction appeared to be weak because we did not observe a sigmoidal transition in the heat effects [33], we note that non-sigmoidal titrations may not be indicative of weak binding in complex systems such as LPS membranes. Sigmoidal binding curves are expected at sufficiently high *c*-values [34] for interactions where the multiple independent binding sites (MIS) model applies [33,35]. Here, the MIS model is unlikely to apply because binding to aggregated LPS molecules [36,37] likely leads to changes in the aggregate morphology [38,39]. We, therefore, did not expect sigmoidal binding curves but rather gradually decreasing binding curves or complex binding curves. This expectation is born out, for example, in studies where non-specifically binding anti-microbial peptides are titrated into LPS solutions and *K*d is > 1 μM. In such studies, sigmoidal ITC curves are not usually observed and in the few cases where curves display sigmoidal-like patterns, the MIS model is unlikely to be applicable [39,40,41,42,43,44,45,46,47]. Attempts to obtain a sigmoidal curve for the present system would also require increasing LPS concentrations, which may not be feasible here [29]. Thus, while quantitative *K*_d_ values were not determined, clear interaction evidence between OligoG and Ca^2+^-mediated LPS exists, and the ITC data can serve for qualitative comparisons.

In the presence of 1 mM Ca^2+^, the heat effects of OligoM with LPS did not change significantly from the combined heat effects for the reference dilution experiments, suggesting that at low calcium concentrations, OligoM and LPS did not interact. Even in the presence of 2.5 mM Ca^2+^, the absence of clear differences in heat effects of diluting OligoM and titrating OligoM into LPS suggests a very weak binding interaction at most (Figure 4). Considering that interactions of OligoG and OligoM involve the same functional groups, enthalpies of interaction for both OligoG and OligoM are expected to be of the same order of magnitude. Given the heat effects observed for OligoG, it is unlikely that the binding of OligoM is masked by a low enthalpy of interaction. Thus, the ITC data implies that, in the presence of Ca^2+^, OligoG binds more strongly to LPS than OligoM.

### 3.3. The Antibiofilm Effects of Low MW Alginate Oligosaccharides

The dose-dependent reduction in bacterial growth was evident during exponential growth for both OligoG and OligoM (Figure 5A), which was significant at 6% between 16 and 40 h compared to the untreated control (MSD = 0.11). At stationary phase (where the line graph plateaus) this effect was lost, although both 6% OligoG and OligoM continued to demonstrate reduced growth.

CLSM showed a dose-dependent reduction in *P. aeruginosa* NH57388A biofilm growth in the presence of both OligoG and OligoM at concentrations ≥ 0.5% (Figure 5B). Biofilm inhibition, quantified by COMSTAT image analysis of the CLSM z-stack images, demonstrated a reduction in bacterial biomass volume at ≥2%, which was statistically significant at 6% (*p* < 0.05; Figure 5C).

A low-volume QS assay to record changes in violacein production in reference *C. violaceum* ATCC 31532 and mutant *C. violaceum* CV026 strains was employed to assess the effects of the alginate oligosaccharides on bacterial cell signaling. Highly active, moderately active, and inactive compounds were classed according to their degree of inhibition of violacein production. CV026 produces violacein only in the presence of exogenous AHLs, making it able to differentiate between QS inhibitors (QSIs) and quorum quenchers (QQs) due to direct interference with and degradation of these signaling molecules. The oligosaccharides showed a dose-dependent effect on violacein inhibition (Figure 5D). The lowest concentration of oligosaccharide used (0.5%) showed <40% inhibition and therefore was classed as inactive. At concentrations between 2 and 6%, however, both OligoG and OligoM demonstrated moderate QSI activity, with 6% OligoG showing the strongest inhibitory effect at >90%, classing it as a highly active QS inhibitor. Importantly, all results demonstrated <40% inhibition of bacterial viability, showing that these effects were not solely due to inhibition of cell growth.

### 3.4. Azithromycin Potentiation in the Presence of Low MW Alginate Oligosaccharides

Both the non-mucoid (PAO1) and mucoid (NH57388A) strains of *P. aeruginosa* tested demonstrated weak antibiotic potentiation of azithromycin in the presence of OligoG (resulting in a two-fold and one-fold reduction in MIC, respectively, at 6%). In contrast, OligoM failed to show any synergy with azithromycin (Table 1).

Dual alginate oligosaccharide and azithromycin therapy of *P. aeruginosa* PAO1 biofilms was conducted to determine synergistic growth responses using a cell viability assay. In the absence of any alginate oligosaccharides, 512 µg/mL azithromycin was required to completely eradicate the biofilm (Figure 6; half-maximal inhibitory concentration [IC_50_] 315.8 µg/mL). Dual treatment with OligoM at 2% showed no effect (IC_50_ 316.0 µg/mL), while at 6%, cell biomass was significantly reduced (MSD = 85,393), having an IC_50_ of 53.0 µg/mL. In contrast, OligoG significantly potentiated the effect of azithromycin at both 2% and 6% in a dose-dependent manner, with an IC_50_ of 161.8 µg/mL and 85.6 µg/mL, respectively. Furthermore, total biofilm eradication occurred at 256 μg/mL azithromycin with 2% OligoG and 128 μg/mL azithromycin with 6% OligoG.

## 4. Discussion

Structure–activity relationships of several low-MW alginate oligosaccharides have highlighted variations in G-M composition as important determinants of biological activity [1]. The β-1,4-glycosidic bonds of the M-block result in a more extended structure, compared to the helical/axial conformation formed with the α-1,4-glycosidic bonds of the G-block, which form strong hydrogels in the presence of divalent cations [48]. A range of biological effects have been reported for low-MW alginate oligosaccharides, including anti-tumor, antioxidant, anti-inflammatory, anti-microbial, and neuroprotective effects [2,9,15,49,50,51,52]. However, in many of these studies, the precise composition, length, structure, and purity of the alginates used are not consistently defined, which leads to difficulties in translating/interpreting structure–activity relationships for low-MW alginate oligosaccharides.

These studies demonstrate the markedly contrasting Ca^2+^-binding capabilities of the G- and M-block alginates and their effects on *P. aeruginosa* LPS binding. Previous electrophoretic light scattering and atomic force microscopy studies have shown the ability of OligoG to resist hydrodynamic shear when bound to the pseudomonal cell surface [14], a feature also seen with OligoM in the FTIR studies here. Cations contribute strongly to the stability of the bacterial outer membrane and to LPS integrity [53]. G-subunits are known to show preferential binding to Ca^2+^, a property attributed to the egg-box-like tertiary structure of the G-block alginate oligosaccharides [4]. In the MD simulations employing short-chain alginates (DP 6), the clear (and contrasting) involvement of Ca^2+^ binding in the penetration of the G-blocks into the membrane model was apparent. The findings of previous studies, which hypothesized a weak interaction between OligoG and pseudomonal LPS [54], are supported by the ITC results, which showed a clear and dose-dependent interaction with Ca^2+^ between LPS and OligoG, which was absent for OligoM.

Direct and indirect effects of divalent cations within bacterial communities can be seen in attachment and electrostatic interactions and also in proteome responses such as extracellular polymeric substance (EPS) synthesis [55]. Calcium also imparts mechanical strength to biofilms by facilitating cross-linking of the biofilm matrix [53]. In light of the cooperative processes involved in alginate gelation [32], it is reasonable to assume that these interactions are very sensitive to Ca^2+^ concentration but also to structural differences in the alginates, such as the differences between OligoG and OligoM. Recent studies employing EPS labeling and MD simulations suggested a possible, direct role for OligoG in disrupting the Ca^2+^–DNA networks present in biofilms, thereby facilitating biofilm disruption [15]. OligoG has previously been shown to reduce bacterial growth in a dose-dependent manner for a range of pathogens [13], properties which were also evident for the M-block alginates in this study with *P. aeruginosa*. Although previous studies have demonstrated that OligoM and OligoMG do not exhibit the same antibiofilm activity as OligoG CF-5/20 [13], other studies employing M- and G-block fractions (<5 kDa) have demonstrated some (albeit limited) antibacterial activity against both Gram-positive and Gram-negative marine pathogens [56].

The effect of OligoM was evident here in both planktonic bacteria and bacterial biofilms. Whereas bacterial colonization is growth-dependent, bacterial biofilm attachment, mediated by electrostatic interactions and EPS production (quantifiable as bacterial biomass), is a Ca^2+^-dependent process. These observations indicate that there are different mechanisms of action responsible for these effects, with sequestration of divalent cations at the bacterial cell membrane favored by G-blocks, while the electrostatic interactions common to both G and M-blocks are responsible for biofilm disruption effects. OligoG has previously been shown to modify bacterial QS signaling [14,57], a finding confirmed by the highly effective QSI effect of OligoG seen here, compared to the moderate QS inhibition activity demonstrated by OligoM.

Although a range of biological effects for alginate oligosaccharides and the biosafety of M- and G-block structures have been shown, very few products have reached clinical trials. A low-MW mannuronic acid oligosaccharide, sodium oligomannate (GV-971) developed in China, appears to show improved cognitive function in mild to moderate Alzheimer’s disease after oral administration [48]. While there is a healthy skepticism about the validity of those studies [58], it does highlight the increasing interest in the gut–brain axis and the possibility of treating neurological disorders by altering the gut microbiota with alginate [59]. An alginate oligosaccharide with high G-block content > 85% and DPn 19 (OligoG CF-5/20) is in clinical development for the management of cystic fibrosis (CF) and has demonstrated an excellent safety profile and high tolerability in clinical trials (NCT00970346; NCT01465529; NCT03822455; NCT02157922; NCT02453789) [60]. People with CF rely heavily on the chronic use of antibiotics, resulting in selective pressure of MDR bacterial pathogens within the lung [61]. The pathoadaptive mutations that occur affect motility, QS signaling, and antibiotic resistance [61]. OligoG CF-5/20 has previously been shown to potentiate the activity of anti-microbial agents against both in vivo and in vitro antibiotic-resistant bacterial biofilms [12,13]. In this study, OligoG showed potentiation of azithromycin against both non-mucoid and mucoid pseudomonal strains but with an enhanced anti-biofilm activity compared to OligoM. Interestingly, when biofilms were chronically exposed to the macrolide azithromycin, OligoG CF-5/20 has also been shown to reduce the cross-resistance of *P. aeruginosa* to other antibiotic classes, such as the monocyclic beta-lactam aztreonam [62]. The ability of OligoG to sustain its antibiotic potentiation properties in these biofilm models highlights its potential utility in treating biofilm-related infections.

The ability to generate alginate oligosaccharide structures of defined G:M composition and DPn (length) affords the opportunity to design alginate formulations for specific clinical applications, whether that is related to biofilm disruption, potentiating the effects of antibiotics/antifungals or facilitating drug delivery across mucosal barriers. Confirming such structural activity relationships, alongside a greater understanding of epimerase and lyase enzymology in alginate production, continues to broaden the opportunities available for tailoring products for specific patient needs in personalized medicines [19].

## 5. Conclusions

This study focused on the structure–activity relationship of M- and G-block alginate oligosaccharides with comparable DPn currently being developed for anti-microbial applications. These experiments demonstrated that both OligoG and OligoM led to a reduction in bacterial growth and disruption of biofilm formation, which were not solely related to Ca^2+^ binding ability. Interestingly, the G-block oligosaccharides interacted most strongly with the LPS of Gram-negative bacteria and exhibited a far stronger QSI effect than OligoM. Synergistic responses for total biofilm eradication with azithromycin were also shown to be superior for OligoG compared to OligoM in biofilm models. This study confirms the potential clinical utility of G-block alginate oligosaccharides in treating infections of MDR bacterial pathogens.

## Figures and Tables

**Figure 1 biomolecules-13-01366-f001:**
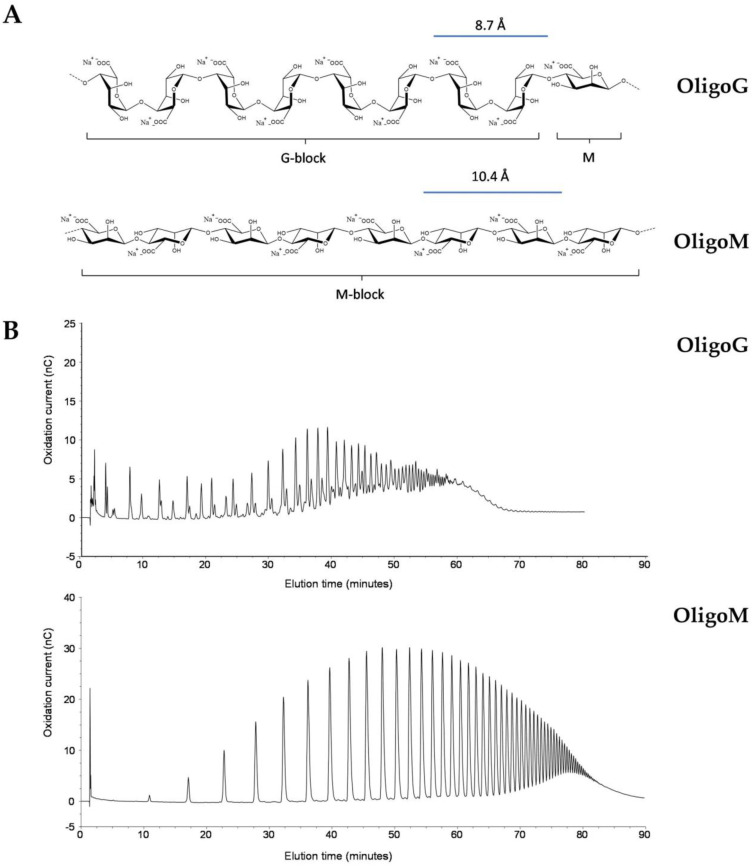
Structure and characterization of alginate oligosaccharides OligoG and OligoM. (**A**) Structures of α-L-guluronate (>85% in OligoG) and β-D-manuronate (100% in OligoM). (**B**) Characterization of OligoG and OligoM with high-performance anion-exchange chromatography with pulsed amperometric detection (HPAEC-PAD).

**Figure 2 biomolecules-13-01366-f002:**
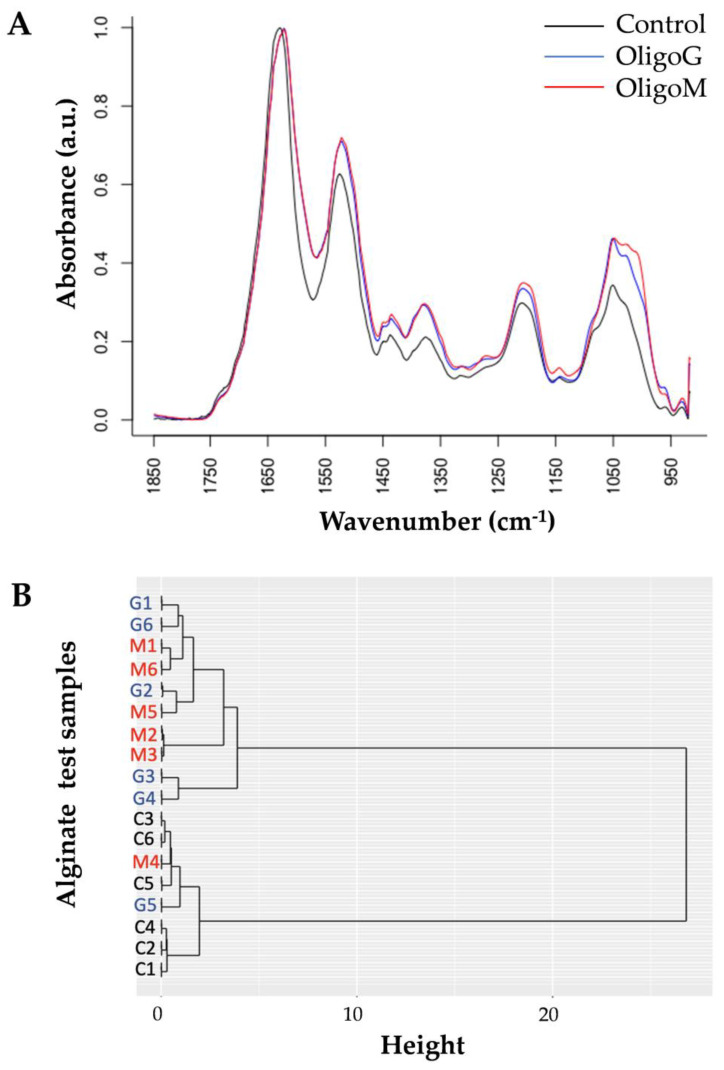
Fourier-transform infrared spectroscopy (FTIR) of alginate-treated *P. aeruginosa* (PAO1) cells. (**A**) FTIR absorbance spectra showing differential peak positions between control (black) and 0.5% OligoG (blue) or 0.5% OligoM (red)-treated cells following hydrodynamic shear. (**B**) Hierarchical cluster analysis (HCA) grouping of the 18 spectra: control C1-C6 (black), OligoG G1-G6 (blue), OligoM M1-M6 (red), and n = 6 per biological replicate.

**Figure 3 biomolecules-13-01366-f003:**
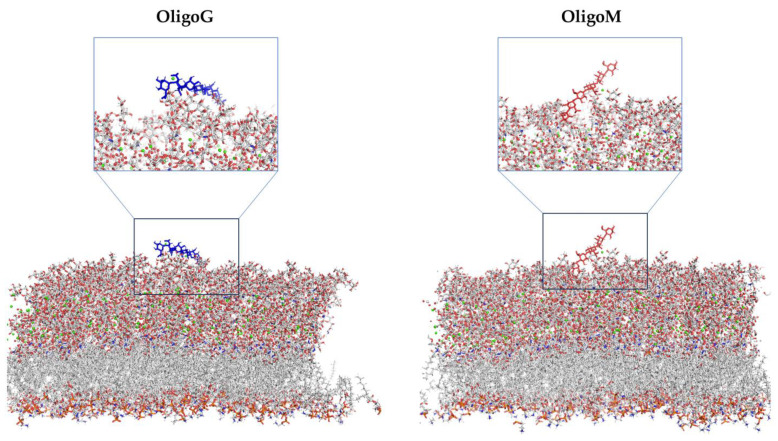
MD simulations of the LPS-DPPE bilayer membrane with G-block and M-block DP 6 taken as a mean from the simulation of 300 ns (calcium molecules highlighted as orange circles).

**Figure 4 biomolecules-13-01366-f004:**
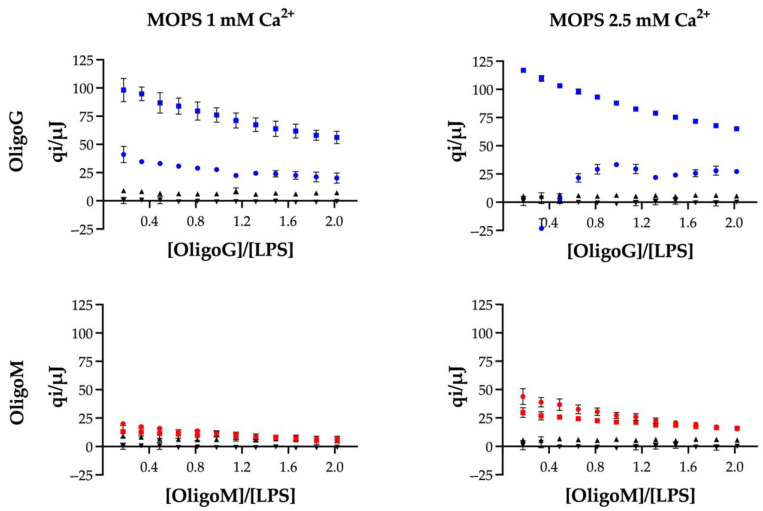
Interaction between OligoG (blue) or OligoM (red) and pseudomonal LPS. Heat effects per injection (qi) for the titration of 20 mg/mL alginate oligosaccharide into 10 mg/mL LPS (□), 20 mg/mL alginate oligosaccharide into buffer (○), buffer into 10 mg/mL LPS (Δ), and buffer-into-buffer (∇) at 37 °C. Buffer: 100 mM NaCl and 20 mM MOPS (+ NaOH adjusted to pH 7) with CaCl_2_ (1 mM or 2.5 mM final free Ca^2+^).

**Figure 5 biomolecules-13-01366-f005:**
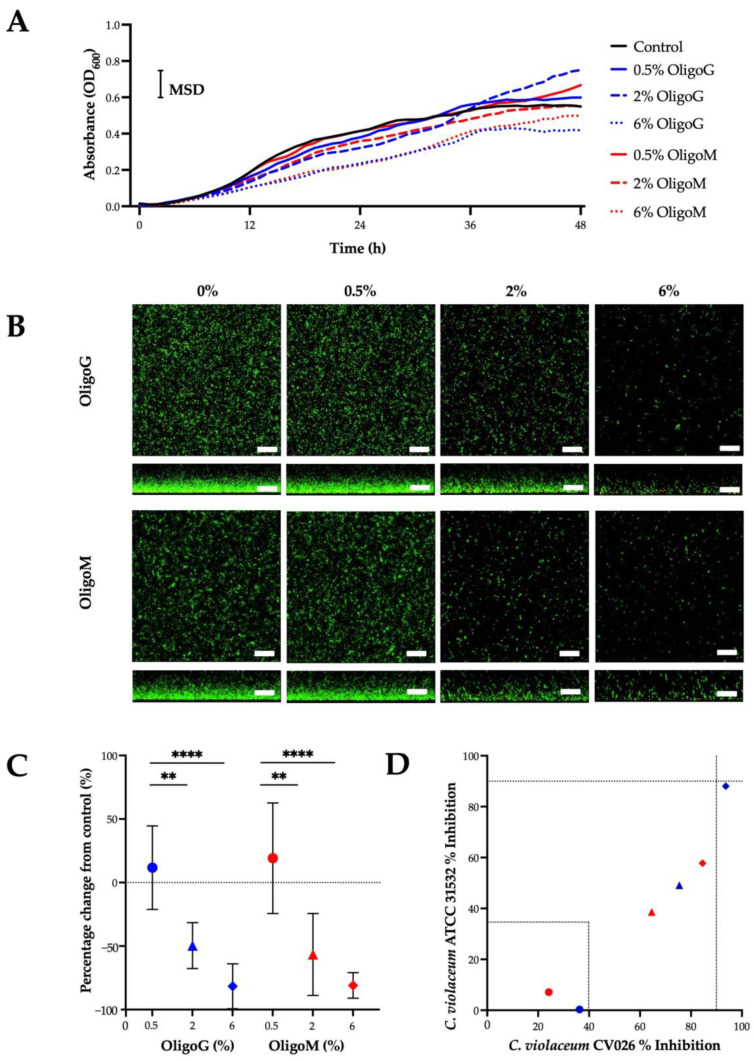
Effect of alginate oligosaccharides on pseudomonal growth. (**A**) Growth curves of *P. aeruginosa* (NH57388A) ± 0–6% OligoG (blue) or OligoM (red). (**B**) Growth of *P. aeruginosa* NH57388A biofilms (24 h) in 0–6% oligosaccharide visualized by CSLM of LIVE/DEAD staining (green, live; red, dead; scale bar 30 µm) with (**C**) corresponding COMSTAT analysis (** *p* < 0.01, **** *p* < 0.0001). (**D**) Inhibition (%) of violacein production in *C. violaceum* ATCC 31532 and *C. violaceum* CV026 by 0.5% (○), 2% (Δ), or 6%(◊) OligoG (blue) or OligoM (red).

**Figure 6 biomolecules-13-01366-f006:**
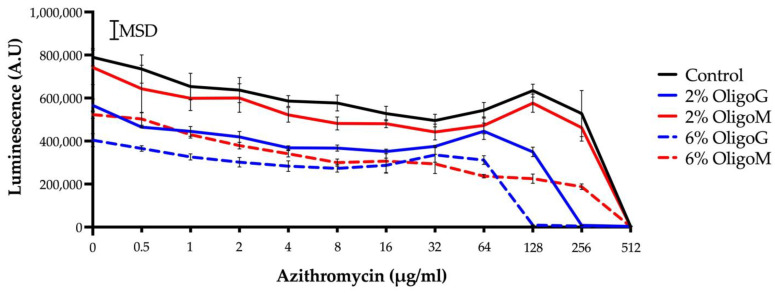
The effect of OligoG (blue) and OligoM (red) at 0–6% on *P. aeruginosa* (PAO1) biofilms ATP production (19 h) in the presence of azithromycin (0.5–512 µg/mL), using the BacTiter-Glo™ Microbial Cell Viability Assay.

**Table 1 biomolecules-13-01366-t001:** MICs of azithromycin (AZM) alone and with increasing concentrations of OligoG or OligoM (2% and 6%) against *P. aeruginosa* (PAO1) and (NH57388A).

		AZM MIC * (µg/mL) at Indicated Oligosaccharide Concentration (%)		
		0	2	6
OligoG	*P. aeruginosa* (PAO1)	128	64	32
	*P. aeruginosa* (NH57388A)	32	32	16
OligoM	*P. aeruginosa* (PAO1)	128	128	256
	*P. aeruginosa* (NH57388A)	32	32	32

* AZM, azithromycin; MIC, minimum inhibitory concentration. The shaded area represents significant potentiation of antibiotics with increasing OligoG concentration.

## Data Availability

The data presented in this study is openly available in Zenodo at doi:10.5281/zenodo.8317059.

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
