# Peer review of "Structure–Activity Relationships of Low Molecular Weight Alginate Oligosaccharide Therapy against Pseudomonas aeruginosa"

_biomolecules, 2023, doi:10.3390/biom13091366_

Round 1
Reviewer 1 Report
Authors present application of the low molecular weight alginate oligosaccharides for interference with bacterial cell walls and biofilm formed by P. aeruginosa (QS) along with mechanistic details concerning the role of Ca(II) cation in this process. For this reason the studies involved hydrolysis and HPLC separation of the two fractions with NPs=19, being isomers on the glycosidic bonds (oligoG and and oligoM) and therefore, having different Ca(II) binding capabilities. Authors used several experimental techniques and molecular dynamic calculations to confirm that oligoG exhibited stronger interactions with bacterial LPS and a stronger QS inhibitory effect than oligoM. Interestingly,
oligoG showed an increased potentiation of the antibiotic azithromycinin min. inhibitory concentration.
Since numerous studies showed that alginate saccharides are active against multiresistant strains, subject is important for contemporary medicine and should attract wide readership.
Article is well written but needs several minor corrections or amendments as shown below:
1.Why compounds with Nps = 19 were used in this research?
2. Although biological source of the low molecular weight saccharides rich in M- or G-form was properly chosen, hydrolysis in case of oligo-G-form may give a variety of polymers. Please mark what fraction shown on chromatograms (Fig. 1B) was selected in these studies. Provide for this fraction mass spectrometry data confirming monodispersity and molecular mass?
3. AMBER is a name of a program and should be written in capital letters.
4. Molecular weight is usually denoted as “MW”.
Moreover:
5. P. 4/152: “...at 10 mg/ml,i.e.~0.5 mM,and OligoG and OligoM at 20 mg/ml,i.e.~6.25 mM,were loaded into the calorimeter sample cell an…”
What molecular mass was used for this approximation?
6. P. 6/Figure 1: give description of the values shown in x,y axes.
7. P. 8/Fig. 3: What is the number of monomers used in MD?
8. P. 12/ 373: “ azithromycin therapy of P. aeruginosaPAO1 bio…”
Bacterial species should be written in italics.
Summing up this is interesting article with clear message, that is supported by adequate eperimental and in silico data and might be published with minor revision.
Only small typing errors shouls be corrected.
Reviewer 2 Report
The idea is publication-worth, the paper is well written, experiments are well done and data are well discussed. Authors made an excellent work and used various approaches to characterize the effects of two alginate oligosaccharides on P. aeruginosa cells, including the biofilm embedded cells. All obtained data are relevant and clearly show the antimicrobial potential of these compounds. Nevertheless, some issues should be addressed before considering for publication
Major
ITC. Please provide a classical titration curves for these measurements and show the KD for each interaction. See other papers which uses ITC to characterize the interaction between macro molecules and ligands.
For the best of my experience in ITC, sometimes the heat release is low because of side effects and the KD seem the best feature of interaction, not the release of the heat.
It should be notes that for OligoG no enough saturation is obtained , the final signal should be less than 50 % of initial, and more injections are obviously required.
If the KD values will be calculated from these data, it could be that OligoM has higher affinity. This will also fit with thiner biofilm shown on Z stack panels in fig 5B in samples treated with OligoM. Of note, it seems strange that biomass quantities in biofilms treated with either Get or M are similar in fig 5C, please check your calculations.
Table 1. In series of two fold dilutions, a significant difference can be suggested if 4x change in mic is observed, ie 2 dilution steps
Figure 6. When assessing with vital tests the viability of biofilms with different initial density this is more clear to compare the IC50 values, ie the concentration providing 2 fold reduction of viability, or IC90, 10 fold reduction.
Minor
Line 184 please specify what furanone has been used
Line 210 Please add a reference to some works where the azitromycin is recommended to treat Pseudomonas infections
Figure 1, it would be nice to make an X axes of the same scale, i.e. 1-95 min to compare the size distributions
Round 2
Reviewer 2 Report
While authors gave valuable answers to reviewer's questions, some issues remain to be fixed.
1) ITC. Please provide a classical titration curves for these measurements and show the KD for each interaction. See other papers which uses ITC to characterize the interaction between macro molecules and ligands. See https://www.cell.com/fulltext/S0092-8674(05)00503-9 Fig 2.
To calculate KD, author must firstly evaluate the stoichiometry of interaction from the curves and then use one-binding site or multiple-binding site models to calculate KDs
Without these figures and KD values this section in incomplete.
2) The biomass quantities in biofilms treated with either Get or M are similar in fig 5C, while CLSM figure show that in M-treated the biomass is less. I realize that in series of experiments biofilms are different, it is clear from images with 0%. Therefore I suggest to normalize data in Fig 5C to control well, and show the residual biomass in % to avoid misunderstanding.
3) Figure 6. When assessing with vital tests the viability of biofilms with different initial density this is more clear to compare the IC50 values, ie the concentration providing 2 fold reduction of viability, or IC90, 10 fold reduction.
As stated in lines 225-227, the inoculum for biofilm growth was standardised at the start of the experiment to avoid any inconsistencies in the initial biofilm density: “Overnight cultures of P. aeruginosa (PAO1) in TSB were adjusted to an OD600 of 0.005 in MH broth (LabM) before adding 30 µl of inoculum to the wells of 96-well black plates.”
The figure lacks the viability data for controls, ie wells without any treatment, and treated with only Oligos.
